# The Efficacy and Safety of Immune Checkpoint Inhibitors in Adrenocortical Carcinoma: A Systematic Review and Meta-Analysis

**DOI:** 10.3390/cancers16050900

**Published:** 2024-02-23

**Authors:** Obada Ababneh, Alina Ghazou, Mohmmad Alawajneh, Saleh Alhaj Mohammad, Abdullah Bani-Hani, Nasr Alrabadi, Aditya Shreenivas

**Affiliations:** 1Faculty of Medicine, Jordan University of Science and Technology, Irbid 22110, Jordan; aaghazou19@med.just.edu.jo (A.G.); mmalawajneh183@med.just.edu.jo (M.A.); soalhajmohammad187@med.just.edu.jo (S.A.M.); aabanihani189@med.just.edu.jo (A.B.-H.); 2Department of Pharmacology, Faculty of Medicine, Jordan University of Science and Technology, Irbid 22110, Jordan; nnalrabadi@just.edu.jo; 3Department of Hematology and Oncology, Medical College of Wisconsin, 8701 Watertown Plank Road, Milwaukee, WI 53226, USA

**Keywords:** adrenocortical carcinoma, immune checkpoint inhibitors, immunotherapy, meta-analysis

## Abstract

**Simple Summary:**

Adrenocortical carcinoma (ACC) is a rare and aggressive cancer. There are few treatment options for ACC. Immune checkpoint inhibitors (ICIs) have revolutionized the treatment of different cancers. In this systematic review and meta-analysis, we reviewed studies that used ICIs in the treatment of ACC. We found that the use of ICIs has a modest efficacy but a good safety profile in ACC. In addition, the use of either single-agent ICIs or ICI combinations did not differ in terms of the tumor response. The use of ICIs also showed promising overall survival benefits. We recommend conducting more studies to determine the best candidates for ICI treatment in ACC.

**Abstract:**

Immune checkpoint inhibitors (ICIs) have revolutionized the treatment of different malignancies. However, their efficacy in advanced adrenocortical carcinoma (ACC) remains uncertain. Thus, we conducted a systematic review and meta-analysis to summarize the efficacy and tolerability of ICIs in patients with advanced ACC. We searched PubMed, Scopus, and CENTRAL for studies that used ICIs in ACC. Studies with more than five patients were included in the meta-analysis of the objective response rate (ORR), disease control rate (DCR), overall survival (OS), progression-free survival (PFS), and grade 3/4 adverse events. Twenty studies with 23 treatment arms and 250 patients were included. Single-agent anti-PD1 or anti-PD-L1 treatment was utilized in 13 treatment arms, whereas an anti-PD1 or anti-PD-L1 and anti-CTLA4 combination was used in 4 treatment arms. Other anti-PD1- or anti-PD-L1-based combinations were used in five treatment arms. The ORR was 14% (95% CI = 10–19%, I^2^ = 0%), and the DCR was 43% (95% CI = 37–50%, I^2^ = 13%). The combination anti-PD1- or anti-PD-L1-based treatment strategies did not correlate with higher responses compared with monotherapy. The median OS was 13.9 months (95% CI = 7.85–23.05), and the median PFS was 2.8 months (95% CI = 1.8–5.4). ICIs have a modest efficacy in advanced ACC but a good OS. Further studies are needed to investigate predictive biomarkers for ICI response and to compare ICI-based strategies with the current standard of care.

## 1. Introduction

Adrenocortical carcinoma (ACC) is a rare and highly aggressive tumor arising from the cortex of the adrenal gland [1]. The 5-year survival rates for stage I and stage II are 80% and 60%, respectively. Unfortunately, the detection of ACC in the early stages is not common, and it is usually detected in rather advanced stages, where the 5-year survival rate is less than 15% among patients with metastatic disease [2]. Surgery is the only curative approach to treating localized ACC, but the recurrence rates remain high despite optimal resection [3,4]. Currently, the treatment of metastatic ACC is based on mitotane, which inhibits adrenal steroid production and has a cytotoxic effect on the cancerous adrenal cells. It is the only drug approved by both the US Food and Drug Administration (FDA) and the European Medicine Agencies (EMA) for the treatment of ACC [5]. A combination of mitotane with Etoposide, Doxorubicin, and Cisplatin (EDP) showed a better progression-free survival (PFS) and response rate compared to mitotane and streptozocin in the First International Randomized Trial in Locally Advanced and Metastatic Adrenocortical Carcinoma Treatment (FIRM-ACT) study, the first ever randomized controlled trial on ACC with a large sample size of 304 subjects [3,6]. The median PFS was significantly longer in patients receiving mitotane with EDP compared to patients treated with mitotane and streptozocin (5 vs. 2.1 months, *p* < 0.001). Interestingly, there was no significant difference in the overall survival (14.8 months vs. 12.0 months, respectively) between the two treatment arms. Both regimens had similar serious adverse event rates and a similar quality of life. Thus, mitotane with EDP became the standard of care in patients with extensive disease ineligible for surgery [7].

ACC is sometimes associated with Lynch syndrome, the presence of one or more mutations in DNA mismatch repair genes [8]. Mutations in these genes have been linked to a superior response to immune checkpoint inhibitors (ICIs) in multiple carcinomas [9,10]. ICIs enhance the immune system to kill tumor cells by blocking the interaction between immune checkpoints, namely PD-1 and CTLA-4, and their ligands. ICIs are being used or investigated in various tumors either as single agents or in combination with other treatments, such as chemotherapy, radiotherapy, or targeted therapy. Figure 1 illustrates the immune microenvironment of ACC and its interaction with some of the other targeted therapies, including lenvatinib and mitotane [11,12,13]. Despite the efficacy of ICI and ICI-based combination therapies, their role in rare tumors like ACC is heterogeneous and less well established [14]. For example, ICIs have been effective in Merkel cell carcinoma and cutaneous squamous cell carcinoma, while they have not been as effective in low-grade neuroendocrine tumors [14,15]. In ACC, Raj. et al. conducted a phase II clinical trial of pembrolizumab in 39 ACC patients, with an impressive median overall survival of 24.9 months [16]. Out of the nine patients that had a meaningful cancer response to pembrolizumab, seven patients had a microsatellite-stabile disease. On the other hand, Le Tourneau et al. investigated the use of avelumab, an anti-PD-L1 treatment, in 50 patients with ACC [17]. The median overall survival was 10.6 months with only three patients achieving a partial response as their best response. In previous trials, FDA-approved biomarkers of immunotherapy such as PD-L1 levels, tumor mutation burden (TMB), and microsatellite stability status were not predictive of the ICI response.

Unfortunately, there are few treatment options, variable responses, and a scarcity of randomized clinical trials in this disease space; hence, there is an urgent need to find novel therapeutic options for these patients. Considering all of these points, we conducted a systematic review and meta-analysis to assess the efficacy and safety of using immune checkpoint inhibitors in patients with ACC.

## 2. Methodology

### 2.1. Search Strategy

This study protocol was registered via PROSPERO (ID: CRD42022303733). This study was performed according to the Preferred Reporting Items for Systematic Reviews and Meta-Analyses (PRISMA) guidelines. The PubMed, Scopus, and CENTRAL electronic databases were searched for articles published from their inception up to 28 September 2022 and later updated to up to 6 October 2023. In addition, we searched the ASCO annual meeting library (2014–2022), ESMO annual congress and special meetings (2014–2022), and the European Network for the Study of Adrenal Tumors (ENSAT) 2022 meeting. The references and new citations of the included studies were also screened to ensure no missed studies. The primary endpoints were the objective response rate (ORR), disease control rate (DCR), overall survival (OS), and progression-free survival (PFS). The secondary endpoints were adverse events. The main search terms used in the databases search were the following: “adrenocortical”, “immune checkpoint inhibitors”, “ipilimumab”, “nivolumab”, “pembrolizumab”, “avelumab”, “cemiplimab”, “durvalumab”, “tremelimumab”, “atezolizumab”, “spartalizumab”, “immunotherapy”. The full search algorithms are reported in Appendix A. Two reviewers independently appraised the eligibility criteria and extracted the data.

### 2.2. Eligibility Criteria

We included case reports, case series, retrospective studies, cohort studies, and clinical trials. Studies had to report at least one primary outcome to be included. Non-English studies and studies that used radiotherapy as part of an immunotherapy combination were excluded. Studies with less than 5 patients were excluded from the meta-analysis.

### 2.3. Quality Assessment

Two reviewers independently assessed the risk of bias in each study. Discrepancies between the reviewers’ judgments throughout the process were resolved according to discussions with a third reviewer. The methodological index for non-randomized studies (MINORS) tool was used for case series, non-randomized controlled trials, and single-arm clinical trials [18]. The MINORS evaluation criteria items that we used to assess the quality of the studies included (1) a clear aim, (2) the inclusion of patients in a consecutive manner, (3) prospectively collected clinical data, (4) appropriate endpoints, (5) unbiased assessment of endpoints, (6) an appropriate follow-up time, (7) less than 5% loss during follow-up, and (8) calculation of the study size before study initiation. In the case of single-arm studies, the following items were not applicable: (1) a suitable control arm, (2) contemporary groups, (3) baseline similarity between arms, and (4) appropriate statistical analyses. The items were scored 0 if not reported, 1 if reported but inadequate, or 2 if reported and adequate. Thus, a total score would be 16 for single-arm studies and 24 points for comparative studies. We considered scores of 12 points or more high-quality with a low risk of bias; scores between 8 and 12 points at medium risk of bias; and scores of 8 points or less at high risk of bias.

We used the Joanna Briggs Institute (JBI) critical appraisal tools for the case reports [19]. Items in the tool were scored either “yes”, “no”, or “not clear”. All items are shown in the table. The total risk of bias was considered “high-risk” when “yes” was the answer for less than 50% of the questions for a study, “moderate” if “yes” was the answer for 50–69% of questions, and “low-risk” if “yes” answers were more than 70%.

### 2.4. Statistical Analyses

The ORR, DCR, PR, SD, and PD incidence rates were calculated using proportional meta-analysis. The degree of statistical heterogeneity between studies was evaluated using the I^2^ statistic. A fixed-effects model was employed if there was no statistical heterogeneity (I^2^ < 50%) across the studies. A random-effects model was utilized if there was statistical heterogeneity (I^2^ ≥ 50%) between studies. The publication bias was represented using funnel plots and statistically evaluated using both Egger’s and Begg’s tests. The META package from R software version 4.2.1 (https://www.R-project.org (accessed on 13 November 2023)) was used to analyze the data and build the figures. For pooling the OS and PFS, we used the median of medians methods performed using ‘metamedian’ as described by McGrath et al. [20]. A *p*-value < 0.05 was considered statistically significant.

## 3. Results

### 3.1. Search Results and Characteristics of the Included Trials

Our systematic search retrieved a total of 2451 reports. After removing duplicates and checking the titles and abstracts, 25 articles were identified. Among these, 20 unique studies with 22 treatment arms were included (Figure 2). There were eight phase II trials [16,21,22,23,24,25,26,27], two phase I trials [17,28], three retrospective cohorts [29,30,31], one case series [32], and six case reports [33,34,35,36,37,38], with a total of 248 ACC patients. Monotherapy of an anti-PD-L1 or anti-PD1 treatment was used in 12 treatment arms. An anti-PD1/anti-PD-L1 and anti-CTLA4 combination was used in four treatment arms, and one study had most of its patients receiving an anti-PD-L1 or anti-PD1 treatment, while 4% received an anti-PD-L1 or anti-PD1 with anti-CTLA4 combination. An anti-PD-L1 or anti-PD1 treatment was used with other agents in five treatment arms. Since most of the patients in the Remde et al. study [31] received single-agent ICIs, the study was considered a monotherapy ICI study in subsequent meta-analyses. The characteristics of the included studies are summarized in Table 1.

### 3.2. Risk of Bias Assessment

All studies evaluated using the MINORS tool scored > 12 and thus were regarded to be at a low risk of bias except for three studies with a score less than 12, which were regarded as at medium risk of bias. Regarding the case reports, all of them had a low risk of bias according to the JBI tool; Please refer to Appendix A here.

### 3.3. Meta-Analysis

Meta-analysis is a statistical tool that combines the results of multiple studies to come up with an aggregate, more precise value of an effect. In the following sections, we report the results of our meta-analyses for the ORR, DCR, PR, SD, and PD.

#### 3.3.1. ORR and DCR

Eleven studies with more than five patients, totaling 228 patients, reported data on the ORR. Using the fixed-effects model, the ORR was 14% (95% CI = 10–19%, I^2^ = 0%); see Figure 3A. The subgroup analysis did not show a difference between monotherapy and combination therapy (14%, 95% CI = 10–19%, I^2^ = 0% vs. 17%, 95% CI = 7–34%, I^2^ = 17%, respectively). The DCR was reported in 10 trials. The pooled DCR was 43% (95% CI = 37–50%, I^2^ = 13%); see Figure 3B. The DCR was not different between monotherapy and combination therapy (42%, 95% CI = 36–49%, I^2^ = 27% vs. 50%, 95% CI = 33–67%, I^2^ = 0%, respectively). Both the ORR and DCR were similar based on the type of drugs used; see Appendix A.

#### 3.3.2. Best Overall Response

We evaluated each response rate individually; see Figure 4A–C. Interestingly, no patient achieved a complete response (CR). Partial response (PR) and stable disease (SD) were seen in 14% (95% CI = 10–19%, I^2^ = 0%) and 29% (95% CI = 19–41%, I^2^ = 54%), respectively. A total of 115 patients had progressive disease (PD) with a rate of 50% (95% CI = 44–57%, I^2^ = 53%). Whether patients received monotherapy or combination therapy did not change the response rates.

#### 3.3.3. PFS and OS Results

The median OS (mOS) was reported in six arms with 173 patients. The pooled mOS was 13.9 months (95% CI = 7.85–23.05); see Figure 5A. On the other hand, eight arms with 194 patients had a median PFS (mPFS) of 2.8 months (95% CI = 1.8–5.4); see Figure 5B. The time-specific survival rates were mostly not reported or were reported at different intervals. Thus, we only analyzed the 6-month OS and PFS. The 6-month OS was reported in two studies and was 71% (95% CI = 59–82%, I^2^ = 0%); see Appendix A. Meanwhile, the 6-month PFS was reported by three trials and was 21% (95% CI = 14–30%, I^2^ = 0%); see Appendix A.

#### 3.3.4. Adverse Events

The pooled grade 3/4 AEs rate were reported in eight trials and was 22% (95% CI = 9–45%, I^2^ = 60%); see Appendix A. Due to the heterogeneity of reporting AEs, we implemented unweighted pooled statistics for selected AEs, shown in Table 2. Fatigue was the most commonly reported all-grade adverse event in 24.6% of the patients, while elevated liver enzymes/hepatitis was the most common grade 3/4 adverse event.

#### 3.3.5. Publication Bias

Based on the DCR results, we found no evidence of publication bias (Begg’s *p*-value = 0.531 and Egger’s *p*-value = 0.213). However, the results based on the ORR showed the risk of publication bias (Begg’s *p*-value = 0.085 and Egger’s *p*-value = 0.01); see Figure 6A,B. Thus, we conclude that publication bias may influence our results.

## 4. Discussion

The management of advanced/metastatic ACC remains a major challenge, and there is an urgent need for a novel treatment paradigm. ICIs have revolutionized the treatment of different malignancies. To the best of our knowledge, this is the first systematic review and meta-analysis to summarize the efficacy and tolerability of different ICI agents in patients with advanced/metastatic ACC. We found that the ORR and DCR were 14% and 43%, respectively. The use of combination strategies did not correlate with higher responses compared with monotherapy. Although the type of ICI did not significantly affect the response, avelumab had the lowest ORR of 6%. The current standard of care is mitotane and EDP, which has a 23% ORR based on the results of the FIRM-ACT trial. Interestingly, the pooled mPFS of ICIs in our study was numerically inferior to in the FIRM-ACT trial in the second-line setting (5.6 months vs. 2.8 months). However, the mOS of ICIs was similar to that of mitotane and EDP in the FIRM-ACT trial (14.8 months vs. 13.9 months). Although comparing ICI-based strategies with mitotane and EDP needs a well-conducted randomized clinical trial, we believe that ICIs could be used either after mitotane and EDP progression or for unacceptable adverse events. Other ongoing immunotherapy-based trials are summarized in Table 3.

Although we aimed to study predictive biomarkers for ICI response, the scarcity and heterogeneity of reporting for these biomarkers prevented such analysis. In 2017, the FDA approved the use of pembrolizumab for the treatment of microsatellite-instability-high (MSI-H) solid malignancies [39]. Raj et al. found that out of the six patients treated with pembrolizumab, two patients achieved PR and another two achieved SD [16]. However, they found that tumor microsatellite status did not predict response. Interestingly, one patient with MSI-H died just one month after starting pembrolizumab. Similarly, one of the six cases in the Klein et al. study also had rapidly progressive disease and died within 12 weeks of the initiation of ipilimumab and nivolumab treatment [25]. We are not sure whether these patients fit the criteria of hyperprogression phenomena. However, oncologists should suspect hyperprogression phenomena in patients with rapidly progressive disease after ICIs [40].

No included trial found PD-L1 as a predictive biomarker for ACC. In theory, anti-PD-1 treatment can shut down both the PD-1/PD-L1 and PD-1/PD-L2 axes [41]. Clinically, anti-PD-1 treatment shows very similar responses to anti-PD-L1 treatment [42]. The use of avelumab (anti-PD-L1) had the worst ORR and DCR results. It is unclear why avelumab was associated with a poor response compared to other ICIs. A possible hypothesis could be that a high expression of PD-L1 assessed according to immunohistochemistry is only 3–11% while PD-L2 expression is about 44% in these tumors, which may mean that anti-PD-1 treatment may offer better blockage of the PD L1/L2 axis compared to anti-PD-L1 treatment [43,44]. Thus, future studies should also assess the expression and predictive role of PD-L2. In 2020, the FDA approved pembrolizumab for the treatment of high TMB solid tumors [45]. However, Raj et al. and Remde et al. found that TMB did not correlate with response in ACC [16,31]. Mota et al. reported two cases of TMB-H ACC treated with pembrolizumab. The patient who responded had an additional MSH2 mutation while the other who did not respond did not have any known deleterious mutations [33]. One of the cases in the Edenfield et al. study had a PD-L1 expression of 1%, MSS, a low TMB, and no actionable mutations in metastatic ACC, receiving durvalumab with tremelimumab, and had a sustained complete response at 18 months [27]. These results suggest the classical biomarkers of ICI response are of low benefit in ACC.

Cortisol-secreting ACC has been associated with CD8+ T cell suppression and decreased immune cell trafficking and tumor cell recognition [46,47,48]. High CTNNB1 expression of the Wnt pathway has also been linked to higher cortisol levels and immunotherapy resistance [47,49]. Thus, targeting WNT/β-catenin might bypass the adverse cortisol role and enhance immunotherapy response. Several preclinical studies have shown superior ICI responses when combined with WNT/β-catenin inhibitors [50,51,52]. Currently, we are waiting for the results of several early-phase clinical trials using ICI and WNT/β-catenin inhibitor combinations in advanced solid tumors (NCT03447470, NCT02675946, NCT04166721). In the Carneiro et al. study, all four patients with hormone-secreting ACC had a PD [23]. Habra et al. reported seven patients with hormone-secreting disease, with one patient achieving PR and another three achieving SD, while the other three had a PD [22]. Casey et al. reported a case of rapid progression in a 58-year-old female Lynch syndrome patient with cortisol-secreting ACC treated with pembrolizumab [34]. On the other hand, Weng et al. reported sustained stable disease in hormone-secreting ACC for about two years using a unique combination of mitotane, paraplatin, sintilimab, and etoposide [37]. In the Remde et al. study, four patients out of seven patients achieving PR had glucocorticoid-secreting tumors [31]. However, no association between glucocorticoid excess and OS was found. Alam et al. reported an interesting case of non-hormonally active ACC treated with pembrolizumab and mitotane and achieving a complete radiological response [36]. This case suggests a possible synergistic effect between mitotane and pembrolizumab even in the settings of hormonally inactive disease. In addition, many patients receive exogenous glucocorticoids as a supplement especially if they are being treated with mitotane. This might partially explain the global low response to ICIs. No study detailed patients’ glucocorticoid supplements. Although discounting these supplements is not realistic, dose modifications might be considered when using ICIs to balance the vital bodily need for glucocorticoids and effective disease control using ICIs.

In a systematic review and meta-analysis conducted by Zhou et al., the occurrence of adverse events was associated with a better ICI response [53]. Raj et al. noticed that all the responders in their study also had immune-related hepatitis [16]. Similarly, Klein et al. reported that all patients who achieved disease control from ICIs also had autoimmune hepatitis [25]. Importantly, mitotane can also cause elevated liver enzymes, and monitoring the LFT in patients receiving either mitotane, ICIs, or a combination of both is necessary [40]. Due to the typically limited sample size in ACC trials, it is difficult to investigate such a hypothesis.

Our study has several limitations. First, there is a lot of heterogeneity between studies in terms of the study designs and treatment strategies. Second, the number of patients was small. Third, publication bias was detected in studies reporting the ORR. Lastly, many studies did not report the OS or PFS or on possible predictive biomarkers such as PD-L1, TMB, and MSI. Thus, we could not perform a meta-analysis of them.

## 5. Conclusions

In conclusion, our systematic review and meta-analysis showed that immune checkpoint inhibitors (ICIs) have modest efficacy in patients with advanced adrenocortical carcinoma (ACC), with an ORR of 14% and a DCR of 49%. The use of combination strategies did not correlate with higher responses compared with monotherapy. Although the type of ICI did not significantly affect the response, avelumab had the lowest ORR. We could not find any predictive factor for ICI response. ICIs could be used either after mitotane and EDP progression or in patients with poor tolerance to EDP. Designing interventional studies for rare cancers like ACC can be challenging and often below the feasibility threshold of prospective clinical trials. Based on our data, we could postulate that a combination of ICI-based therapies could be a promising therapeutic option in this disease space.

## Figures and Tables

**Figure 1 cancers-16-00900-f001:**
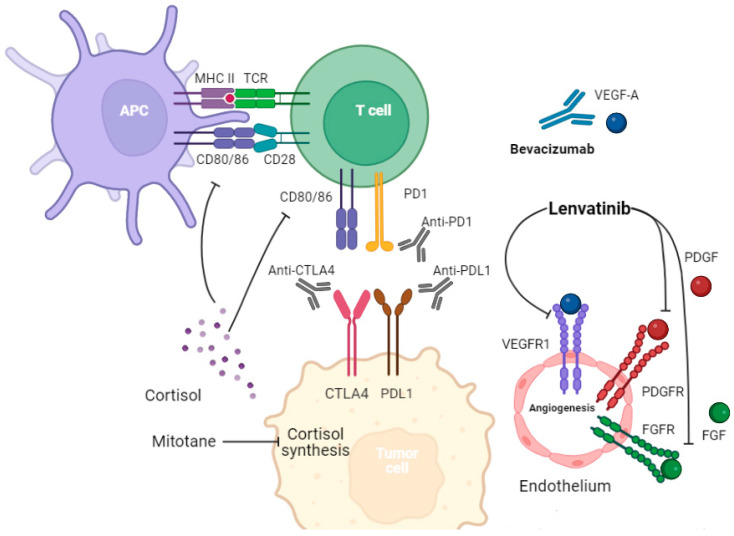
The interaction between immune cells, endothelium, and tumor cells highlights possible treatment strategies. Tumor cells express PD-L1 and CTLA-4, which inhibit the immune system. Endogenous cortisol excess can decrease antigen presentation, inhibit T cell activation, decrease CD8+ T cell infiltration, weaken immune attack, and cause potential immunotherapy failure. Mitotane works by inhibiting the CYP11A1, CYP11B1, CYP11B2, and 3β-HSD enzymes, leading to decreased cortisol secretion. Targeting angiogenesis through multikinase inhibitors can also be helpful in controlling disease. In addition to inhibiting angiogenesis, lenvatinib can alter the immune tumor microenvironment.

**Figure 2 cancers-16-00900-f002:**
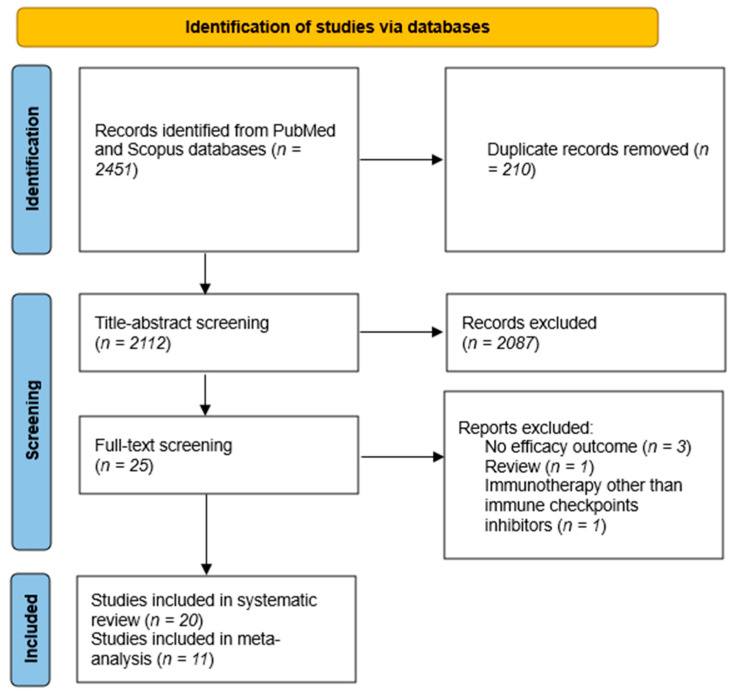
PRISMA flow diagram. After applying our search strategy, 2112 unique reports were found. A total of 20 reports satisfied our eligibility criteria, of which 11 studies had sufficient data to be included in our meta-analysis.

**Figure 3 cancers-16-00900-f003:**
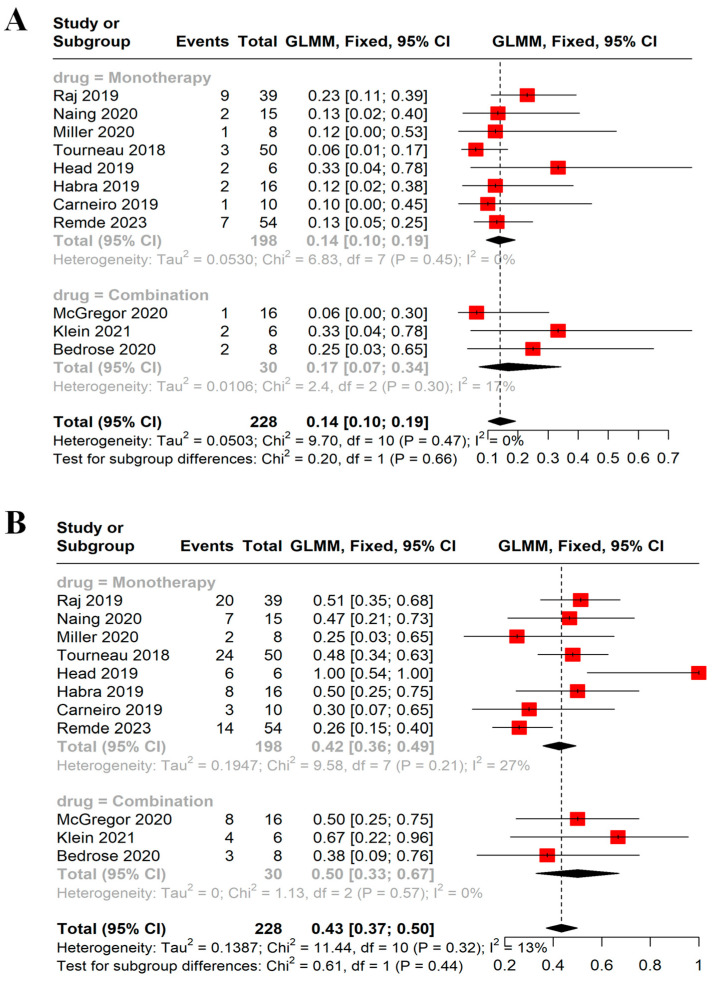
Forest plots demonstrating (**A**) ORR and (**B**) DCR. ORR was 14% with no difference between monotherapy and combination therapy (14% vs. 17%, *p*-value = 0.66). DCR was 43% with no difference between monotherapy and combination therapy (42% vs. 50%, *p*-value = 0.44). Each red square represents the effect size of a study, and the black diamond represents the pooled effect size. ORR: objective response rate, DCR: disease control rate [16,17,21,23,24,25,29,30,31,32].

**Figure 4 cancers-16-00900-f004:**
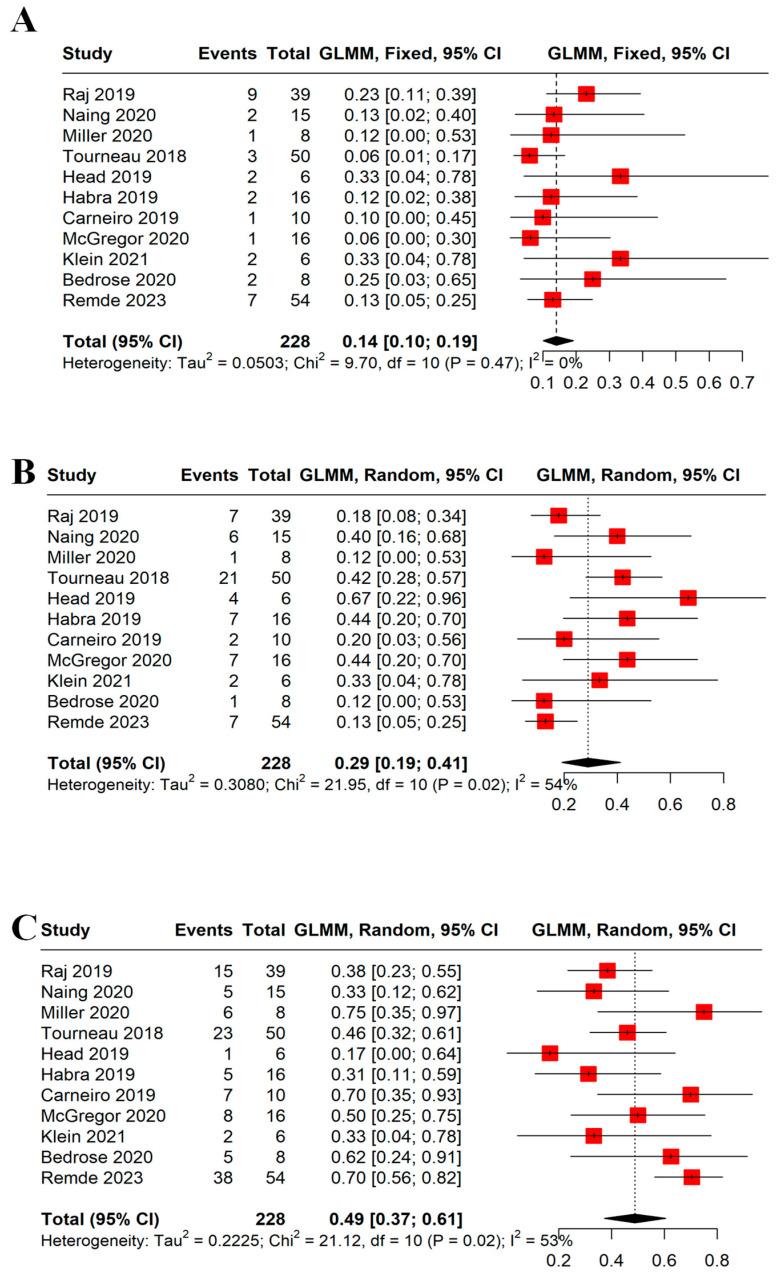
Forest plots demonstrating (**A**) PR, (**B**) SD, and (**C**) PD. The PR, SD, and PD rates were 14%, 29%, and 49%, respectively. A random model was used for SD and PD due to significant heterogeneity (I^2^ > 50%). Each red square represents the effect size of a study and the black diamond represents the pooled effect size. PR: partial response, SD: stable disease, PD: progressive disease [16,17,21,23,24,25,29,30,31,32].

**Figure 5 cancers-16-00900-f005:**
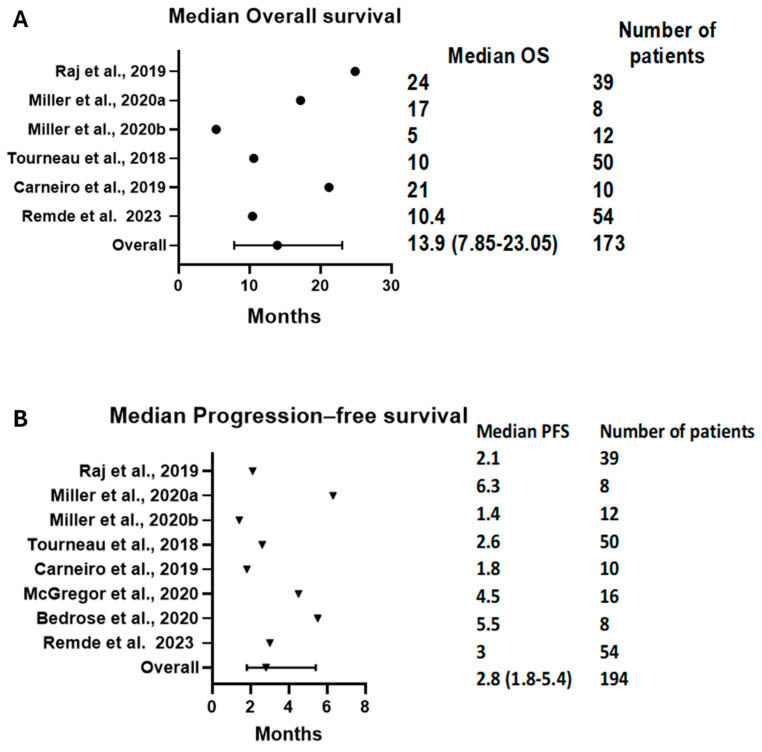
Forest plots demonstrating (**A**) mOS and (**B**) mPFS. The pooled mOS was 13.9 (95% CI: 7.85–23.05), and the pooled mPFS was 2.8 (95% CI: 1.8–5.4). mOS: median overall survival, mPFS: median progression-free survival [16,17,23,24,29,31,32].

**Figure 6 cancers-16-00900-f006:**
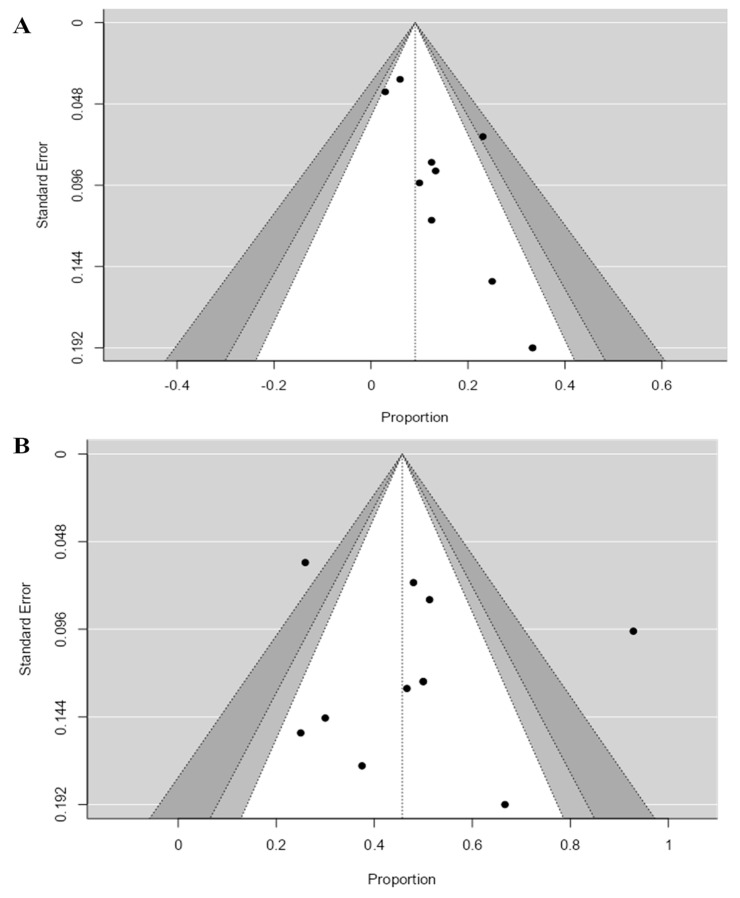
Funnel plots evaluating publication bias of published studies that reported (**A**) ORR and (**B**) DCR. Egger’s test revealed the presence of publication bias in the case of ORR (*p*-value = 0.01) but no publication bias in the case of DCR (*p*-value = 0.213). ORR: objective response rate, DCR: disease control rate.

**Table 1 cancers-16-00900-t001:** Included studies characteristics. *: These studies included multiple tumors, and data regarding ACC were not separated. ^†^: High PDL1 was defined as PD-L1 protein levels of 1% or more. ACC: adrenocortical carcinoma, MSI-H/MMR-D: microsatellite-instability-high/mismatch repair-deficient, NR: not reported.

Study	Study Design	Total Number of Patients (ACC Patients)	Median Age in Years (Range)	Female Percentage	Median Follow-Up Months (Range)	Therapy/Regimen	Stage	ECOG			Cortisol Producing	MSI-H/ MMR-D	Lynch Syndrome	PD-L1 High ^†^
								0	1	2				
Tourneau et al., 2018 [17]	Open-label phase I clinical trial	50 (50)	50 (21–71)	52%	16.5 m (11.7–27.6)	Avelumab (50% of the patients received concurrent mitotane)	Metastatic	19	31	0	NR	NR	NR	12
Raj et al., 2019 [16]	Open-label phase II clinical trial	39 (39)	62 (19–87)	61.5%	17.8 m (5.4–34.7)	Pembrolizumab	Advanced/metastatic	11	28	0	NR	6	2	7
Habra et al., 2019 [22]	Open-label phase II clinical trial	16 (16)	48 (31–78)	50%	NR	Pembrolizumab	Metastatic	1	13	0	10	1	NR	0
McGregor et al., 2020 [24]	Open-label phase II clinical trial	55 (16)	45	72.2%	8.9 m (2.6–17.1)	Nivolumab and ipilimumab	Advanced/metastatic	11	7	0	NR	NR	NR	NR
Naing et al., 2020 [21]	Open-label phase II clinical trial	127 (15)	56 (22–84)	46.5%	NR	Pembrolizumab	Advanced/metastatic	15	112	0	NR	NR	NR	Mixed *
Remde et al., 2023 [31]	Retrospective cohort	54 (54)	46 (19–70)	57.4%	NR	Pembrolizumab: 59% Nivolumab: 24% Avelumab: 11% Atezolizumab: 2% Ipilimumab and nivolumab: 4%	Metastatic	NR			28	3	1	8
Miller et al., 2020 [29]	Retrospective cohort	15 (15)	43 (19–62)	46.7%	83.0 m (22.5–83.0)	Pembrolizumab/MKI	Stage I: 1 patient Stage II: 4 patients Stage III: 4 patients Stage IV: 5 patients	NR			NR	NR	NR	NR
Carneiro et al., 2019 [23]	Open-label phase II clinical trial	10 (10)	57 (31–67)	70%	4.5 m (0.1–25.6)	Nivolumab	Metastatic	5	3	2	4	0	NR	6
Bedrose et al., 2020 [32]	Case series	8 (8)	38 (21–49)	50%	NR	Pembrolizumab and lenvatinib	Stage II: 3 patients Stage III: 2 patients Stage IV: 3 patients	NR			3	0	0	NR
Head et al., 2019 [30]	Retrospective cohort	6 (6)	45 (24–65)	100%	NR	Pembrolizumab and mitotane	Stage I: 1 patient Stage III: 3 patients Stage IV: 2 patients	1	5	0	3	1	2	NR
Klein et al., 2021 [25]	Open-label phase II clinical trial	6 (6)	48 (22–72)	66.7%	14.7 m	Nivolumab and ipilimumab	Stage II: 1 patient Stage III: 3 patients Stage IV: 2 patients	5	1	0	2	2	NR	2
Geoerger et al., 2019 [26]	Open-label phase II clinical trial	154 (4)	13 (8–15)	48.1%	8·6 m (2·5–16·4) [Mixed *]	Pembrolizumab	Mixed *	NR			NR	NR	NR	4
Mota et al., 2018 [33]	Case report	2 (2)	28 and 40	50%	NR	Pembrolizumab	Metastatic	NR			NR	1	0	NR
Edenfield et al., 2021 [27]	Open-label phase II clinical trial	50 (2)	62 (26–78)	56%	NR	Durvalumab and tremelimumab	Advanced/metastatic	Mixed *			NR	0	0	1
Sakamuri et al., 2017 [28]	Open-label phase I clinical trial	36 (2)	56 (19–75)	66.7%	Mixed *	Ipilimumab and lenalidomide	Advanced/metastatic	Mixed *			NR	NR	NR	NR
Casey et al., 2018 [34]	Case report	1 (1)	58	100%	NR	Pembrolizumab	Stage III	NR			1	1	1	0
Caccese et al., 2019 [35]	Case report	1 (1)	29	0%	NR	Pembrolizumab	Metastatic	NR			1	1	0	0
Alam et al., 2021 [36]	Case report	1 (1)	40	100%	NR	Pembrolizumab and mitotane	Stage IV	NR			NR	0	0	0
Weng et al. 2023 [37]	Case report	1 (1)	36	0%	NR	Sintilimab, mitotane, etoposide, and paraplatin	Stage IV	NR			NR	1	0	NR
Charles et al. 2023 [38]	Case report	1 (1)	32	100%	NR	Ipilimumab and nivolumab	Stage IV	NR			0	0	0	NR

**Table 2 cancers-16-00900-t002:** Adverse events in ICI-based regimens. Adverse events reported in at least three studies were only pooled.

	Number of Studies	All Grades	Grade 3/4
Fatigue	7	32/130 (24.6%)	0 (0%)
Nausea/vomiting	6	28/122 (23%)	1/122 (0.8%)
Elevated liver enzymes/hepatitis	9	27/154 (17.5%)	6/154 (3.9%)
Dyspnea	4	5/39 (12.8%)	0 (0%)
Rash	7	17/139 (12.2%)	0 (0%)
Hypothyroidism	5	14/160 (8.8%)	0 (0%)
Anorexia	3	5/61 (8.2%)	0 (0%)
Diarrhea	5	12/157 (7.6%)	1/157 (0.6%)
Pneumonitis	6	12/192 (6.3%)	6/192 (3.1%)
Adrenal insufficiency	5	8/146 (5.5%)	2/146 (1.4%)
Anemia	5	9/167 (5.4%)	2/113 (1.2%)

**Table 3 cancers-16-00900-t003:** Ongoing immunotherapy clinical trials. Trial status as of 29 September 2023. ACC: adrenocortical carcinoma, AEs: adverse events, DLT: dose-limiting toxicity, ORR: objective response rate, OS: overall survival, PFS: progression-free survival.

NCT Identifier	Phase	Treatment	Tumor Type	Status	Endpoints
NCT05563467	II	Pembrolizumab	ACC	Recruiting	ORR, AEs, QoL
NCT04373265	I	Relacorilant with Pembrolizumab	ACC	Active, not recruiting	DLT, PFS, AEs
NCT05634577	II	Mitotane with Pembrolizumab	ACC	Recruiting	ORR, OS, PFS, AEs
NCT04318730	II	Camrelizumab with Apatinib	ACC	Recruiting	ORR
NCT05036434	II	Pembrolizumab with Lenvatinib	ACC	Not yet recruiting	ORR, AEs
NCT06006013	II	Cabozantinib with Atezolizumab	ACC	Not yet recruiting	ORR, OS, PFS, AEs
NCT04187404	I/II	EO2401 with Nivolumab	ACC and malignant pheochromocytoma/paraganglioma	Recruiting	AEs, OS, PFS
NCT02834013	II	Ipilimumab with Nivolumab	Multiple tumors	Active, not recruiting	ORR, AEs
NCT02637531	I	Eganelisib with Nivolumab	Multiple tumors	Active, not recruiting	DLT, AEs, PFS, OS

## Data Availability

The data presented in this study are available upon reasonable request.

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
