# Peer review of "The Efficacy and Safety of Immune Checkpoint Inhibitors in Adrenocortical Carcinoma: A Systematic Review and Meta-Analysis"

_cancers, 2024, doi:10.3390/cancers16050900_

Round 1
Reviewer 1 Report
Comments and Suggestions for Authors
1) I would like to check with the authors what is the purpose of a simple summary and abstract. In my opinion, they must be combined.
2) The purpose of the review paper must be clearly mentioned at the end of the introduction.
3) All of the figures should be independently informative. E.g. Fig 2. Does not have any description.
4) I would like to know whether the authors prepared Table 1 by themselves or if they found it in the literature. E.g. In the first row, they mentioned several patients: 50 followed by Ref 31. If the authors looked through the 10/20/50 reports for 50 patients, they should cite the whole report.
5) In section 3.3 meta-analysis, the authors should describe what is meta-analysis.
6) The quality of Figures 3,4, and 6 is not good. Moreover, a description must be added to the legend of the figures to make them independently understandable.
Author Response
1. I would like to check with the authors what is the purpose of a simple summary and abstract. In my opinion, they must be combined.
Authors` Response: We would like to thank the reviewer for the feedback. The simple summary section was added based on the Cancers Journal guideline. (https://www.mdpi.com/journal/cancers/instructions#front:~:text=for%20further%20information.-,Simple%20Summary,-%3A%20It%20should).
2. The purpose of the review paper must be clearly mentioned at the end of the introduction.
Authors` Response: Thank you for the suggestion. The last sentence of the introduction section: “Considering all of these points, we conducted a systematic review and meta-analysis to assess the efficacy and safety of using immune checkpoint inhibitors in patients with ACC.”
3. All of the figures should be independently informative. E.g. Fig 2. Does not have any description.
Authors` Response: Thank you for the suggestion.
The following description was added to Figure 2: “After applying our search strategy, 2112 unique reports were found. 20 reports satisfied our eligibility criteria of which 11 studies had sufficient data to be included in our meta-analysis.”
The following description was added to Figure 3: “DCR was 43% with no difference between monotherapy and combination therapy (42% vs. 50%, p-value=0.44). ORR: objective response rate, DCR: Disease Control rate.”
The following description was added to Figure 4: “The PR, SD and PD rates were 14%, 29% and 49%, respectively. A random model was used for SD and PD due to significant heterogeneity (I2>50%). PR: Partial response, SD: Stable disease, PD: Progressive disease.”
The following description was added to Figure 5: “The pooled mOS was 13.9 (95%CI: 7.85-23.05) and the pooled mPFS was 2.8 (95%CI: 1.8-5.4). mOS: median overall survival, mPFS: median progression-free survival.”
The following description was added to Figure 6: “The Egger test revealed the presence of publication bias in case of ORR (P-value=0.01) but no publication bias in case of DCR (P-value=0.213). ORR: objective response rate, DCR: Disease Control rate.”
4. I would like to know whether the authors prepared Table 1 by themselves or if they found it in the literature. E.g. In the first row, they mentioned several patients: 50 followed by Ref 31. If the authors looked through the 10/20/50 reports for 50 patients, they should cite the whole report.
Authors` Response: We created Table 1 ourselves using the 20 studies based on the prespecified eligibility criteria. Each row represents a different study with a unique patient population. Studies with more than 6 patients were reported as aggregates rather than individual reports. We tried our best to summarize the studies characteristics in Table 1.
5. In section 3.3 meta-analysis, the authors should describe what is meta-analysis.
Authors` Response: The following was added to the beginning of section 3.3: “Meta-analysis is the statistical tool that combine the results of multiple studies to come with an aggregate more precise value of an effect. In the following sections, we report the results of our meta-analyses for ORR, DCR, PR, SD, and PD.”
6. The quality of Figures 3,4, and 6 is not good. Moreover, a description must be added to the legend of the figures to make them independently understandable.
Authors` Response: We would like to thank the reviewer for addressing this issue. We have improved the quality of the figures and added the descriptions as mentioned in response 3.
Reviewer 2 Report
Comments and Suggestions for Authors
The reviewed article is a systemic review and meta-analysis analyzing the efficiency and safety of immune checkpoints inhibitors in adrenocortical carcinoma. The paper has a clear layout. The introduction provides sufficient background but it should be supplemented with more extensive information on treatment strategies including immunotherapies, while the first paragraph of the introduction could be reduced. Article includes all relevant references. All information is clearly presented. Results are supported by the proper figures. English language and style are fine. The novelty and scientific soundness are average but the quality of presentation and interest to the readers are high. The strong limitation of the work is the small number of patients. In my opinion, the article should be accepted in its present form.
Line 118 - editor mistake
What does the "+" mean in Table 3?
Author Response
1. In The reviewed article is a systemic review and meta-analysis analyzing the efficiency and safety of immune checkpoints inhibitors in adrenocortical carcinoma. The paper has a clear layout. The introduction provides sufficient background but it should be supplemented with more extensive information on treatment strategies including immunotherapies, while the first paragraph of the introduction could be reduced.
Authors` Response: We would like to thank the reviewer for the feedback. We have removed the following sentences:
“ACC is also a highly aggressive tumor that is associated with poor survival rates and has limited therapeutic options [2].”
“It can occur in both the pediatric and adult populations [3].”
“FIRM-ACT was the first-ever randomized clinical trial for ACC.”
“A number of factors have been suggested to be associated with survival in ACC including age at the time of presentation, stage, grade of tumor, cortisol excess, tumor or hormone-related symptoms, Ki-67 score, and mitotic rate. European Network for the Study of Adrenal Tumours (ENSAT) and modified Weiss scores have been utilized for prognostic stratification of ACC [9–12]”.
We added the following sentence to address the role of ICI in ACC: “For example, ICI has been effective in Merkel cell carcinoma and cutaneous squamous cell carcinoma while it has not been as effective in low-grade neuroendocrine tumors [14, 15]. In ACC, Raj. et al. conducted a phase II clinical trial of pembrolizumab in 39 ACC patients with an impressive median overall survival of 24.9 months [16]. 9 patients had a meaningful cancer response in 7 patients with Microsatellite stabile disease. On the other hand, Le Tourneau et al. investigated the use of avelumab, an-ti-PD-L1, in 50 patients with ACC [17]. The median overall survival was 10.6 months with only 3 patients achieving partial response as their best response.”
2. Article includes all relevant references. All information is clearly presented. Results are supported by the proper figures. English language and style are fine. The novelty and scientific soundness are average but the quality of presentation and interest to the readers are high. The strong limitation of the work is the small number of patients. In my opinion, the article should be accepted in its present form.
Authors` Response: We would like to thank the reviewer for the kind words and the consideration to review our work.
3. Line 118 - editor mistake
Authors` Response: We are unsure what mistake you are referring to. We would be happy to correct any spelling and grammar mistakes during the peer review process or proofreading.
4. What does the "+" mean in Table 3?
Authors` Response: It refers to the use of combination therapy. We replace it with the word “with” to avoid any confusion.
Round 2
Reviewer 1 Report
Comments and Suggestions for Authors
The authors answered my questions and the quality of the MS has been improved.